# Different Types of *Hypericum perforatum* cvs. (Elixir, Helos, Topas) In Vitro Cultures: A Rich Source of Bioactive Metabolites and Biological Activities of Biomass Extracts

**DOI:** 10.3390/molecules28052376

**Published:** 2023-03-04

**Authors:** Inga Kwiecień, Natalizia Miceli, Elżbieta Kędzia, Emilia Cavò, Maria Fernanda Taviano, Ludger Beerhues, Halina Ekiert

**Affiliations:** 1Department of Pharmaceutical Botany, Faculty of Pharmacy, Jagiellonian University, Medical College, 9 Medyczna St., 30-688 Kraków, Poland; 2Department of Chemical, Biological, Pharmaceutical and Environmental Sciences, University of Messina, Viale F. Stagno d’Alcontres, 31, 98166 Messina, Italy; 3Department of Bioproducts Engineering, Institute of Natural Fibres and Medicinal Plants—National Research Institute, 71B Wojska Polskiego St., 60-630 Poznań, Poland; 4Foundation “Prof. Antonio Imbesi”, University of Messina, Piazza Pugliatti n. 1 98123 Messina, Italy; 5Institut für Pharmazeutische Biologie, Technische Universität Braunschweig, Mendelssohnstrasse 1, 38106 Braunschweig, Germany

**Keywords:** St. John’s wort, cultivars, agitated culture, bioreactor culture, antioxidant activity, antimicrobial activity, PGRs testing, phenylalanine feeding

## Abstract

Microshoot agitated and bioreactor cultures (PlantForm bioreactors) of three *Hypericum perforatum* cultivars (Elixir, Helos, Topas) were maintained in four variants of Murashige and Skoog medium (MS) supplemented with 6-benzylaminopurine (BAP) and 1-naphthaleneacetic acid (NAA) (in the range of 0.1–3.0 mg/L). In both types of in vitro cultures, the accumulation dynamics of phenolic acids, flavonoids, and catechins were investigated during 5- and 4-week growth cycles, respectively. The contents of metabolites in methanolic extracts from biomasses collected in 1-week intervals were estimated by HPLC. The highest total contents of phenolic acids, flavonoids, and catechins were 505, 2386, and 712 mg/100 g DW, respectively (agitated cultures of cv. Helos). The extracts from biomass grown under the best in vitro culture conditions were examined for antioxidant and antimicrobial activities. The extracts showed high or moderate antioxidant activity (DPPH, reducing power, and chelating activity assays), high activity against Gram-positive bacteria, and strong antifungal activity. Additionally, experiments with phenylalanine feeding (1 g/L) in agitated cultures were performed reaching the highest enhancement of the total contents of flavonoids, phenolic acids, and catechins on day 7 after the addition of the biogenetic precursor (2.33-, 1.73- and 1.33-fold, respectively). After feeding, the highest accumulation of polyphenols was detected in the agitated culture of cv. Elixir (4.48 g/100 g DW). The high contents of metabolites and the promising biological properties of the biomass extracts are interesting from a practical point of view.

## 1. Introduction

One of the best known species worldwide belonging to the *Hypericaceae* family is *Hypericum perforatum* L. (St. John’s wort). *H. perforatum* is a widely distributed, very popular, and highly valued medicinal species in Central Europe, the United States, and beyond [1]. For many years, the herb of this plant (*Hyperici herba*) had the status of a pharmacopoeial plant raw material in European countries [2]. Today, it is authorized for use in both allopathy and homeopathy. This popularity in medicinal applications is due to the very rich chemical composition of *H. perforatum*. The overground parts of this plant are a rich source of naphthodianthrone compounds (e.g., hypericin), monomeric flavonoids (e.g., quercetin derivatives), biflavonoids (e.g., amentoflavone), and procyanidins. Among the phenolic compounds, the important ingredients also include tannins (catechin polymers) and phenolic acids. These constituents are accompanied by essential oil. Flowers of the plant contain hyperforins, whereas roots were found to be a source of xanthones [3,4]. The herb of *H. perforatum* exhibits antidepressant properties and is one of the top-selling antidepressants in the world [5]. It also shows astringent and spasmolytic effects [6], and it accelerates the healing of smaller wounds and burns [4,7]. The main metabolites exhibit antimicrobial activity [8]. Hypericin also causes sensitization to UV radiation of the skin that is used in medicine and the cosmetic industry [9,10].

The increasing demand for the raw material of *H. perforatum* still leads to excessive exploitation of this species in the wild. To ensure the availability of raw material, the species was introduced to field cultivation to increase the yield, unify the plant material, and improve the resistance to diseases and external conditions. St. John’s wort breeding cultivars with specific functional characteristics were also developed for these needs. Some of them, used in this work, are cv. Elixir which is rich in hypericin, cv. Topas which is characterized by a high production of hypericin, rutoside, and hyperoside, and cv. Helos which is highly resistant to fungal infections [1]. Nevertheless, field-grown plants are still affected by a number of physiological and environmental factors [11].

Biotechnological methods are becoming an alternative to conventional crops. The possibility of controlling and stimulating the production of secondary metabolites, especially those that are valuable pharmaceuticals, under in vitro culture conditions inspired research into the biosynthetic potential of *H. perforatum* cells cultured in different in vitro systems. Specific secondary metabolite production under controlled conditions can be achieved using different strategies, such as selection of culture type, optimization of culture conditions, bioreactor engineering, cell line selection, cell immobilization, biotransformation, precursor feeding, elicitation, and metabolic engineering [12]. Even though most of the important secondary metabolites accumulate in the aerial parts of *Hypericum*, three different in vitro culture types of this species are widely used in biotechnological research. These are suspension cultures, shoot cultures, and genetically transformed hairy root cultures. Among them, large-scale shoot cultures may be a promising approach for their successful bioproduction of biomass and metabolites [13,14].

In our laboratory, we focused on the study of the biosynthetic potential of microshoot cultures of St. John’s wort three cultivars: Elixir, Helos, and Topas. Cultures were grown in Linsmaier and Skoog (LS) and Murashige and Skoog (MS) medium supplemented with 0.1 to 3.0 mg/L of 6-benzylaminopurine (BAP) and 1-naphthaleneacetic acid (NAA) (4 medium variants, 3 weeks). It has been demonstrated that the basic composition of the medium and the concentrations of plant growth regulators (PGRs) substantially affect the morphology of biomass, its growth, and the accumulation of phenolic acids and flavonoids [15,16]. The highest amounts of flavonoids were accumulated in the shoots of the Elixir cultivar, while the highest accumulation of phenolic acids was confirmed mostly in cv. Helos. The content of indole compounds in agar cultures was also studied. The dominant metabolite in this group were 5-hydroxy-L-tryptophan (343.2 mg/100 g DW) and serotonin (319.9 mg/100 g DW) detected in cultures of the Helos cultivar [17].

The purpose of the present study was to elaborate and propose the best growing conditions and biotechnological strategy to obtain a high production and accumulation of phenolic secondary metabolites in three cultivars of St. John’s wort. We examined the dynamics of the accumulation of metabolites in the biomass of agitated cultures maintained for five weeks in four variants of MS medium with different concentrations of BAP and NAA to choose the best moment to improve this process. Bioreactor cultures of three cultivars of *H. perforatum* were also tested. They grow for four weeks in temporary immersion PlantForm bioreactors in the same variants of the MS medium as agitated cultures. Extracts with the highest content of phenolic acids and flavonoids were selected for further examination. Their antioxidant potential was tested using three different methods. In addition, their antibacterial and antifungal activities, and safety using the brine shrimp lethality bioassay were examined. Finally, based on our previous experiments with different in vitro cultures, we fed the culture medium with phenylalanine, which is the biosynthetic precursor of phenolic and polyphenolic metabolites.

## 2. Results and Discussion

### 2.1. Dynamics of Metabolite Accumulation in Agitated Cultures

Agitated cultures of *H. perforatum* cultivars were chosen as an experimental model in our previous research [15,16]. Now, we decided to study the dynamics of biomass growth and the accumulation of three groups of metabolites in agitated cultures maintained in four variants of the MS medium with different BAP and NAA content for five weeks, aiming at the selection of the most optimal variant and growth cycle for further research.

The biomass increments ranged from 1.1 to 12.1 times for cv. Elixir, from 2.3 to 11.9 times for cv. Helos, and from 2.1 to 12.0 times for cv. Topas, depending on the duration of the culture cycle and the variant of the MS medium. Based on dry biomass increments, the agitated cultures of the Elixir and Topas cultivars reach their maximum growth in the period between the third and fifth week of cultivation, while the Helos cultivar does so in the fourth week. Only cultures grown in a medium containing 3 mg/L BAP and NAA show slower growth dynamics (Figure 1). In the cultures cultivated in medium variants with a higher content of growth regulators, a breakdown of the growth curve was observed in the fifth week, which may indicate the beginning of the culture’s withering phase. Significant differences in the appearance of the biomass were also visible. With the increase in the content of growth regulators, the foliage became smaller and the biomass more compact and browner, with the predominance of callus tissue (Figure 2). These differences were described by us in earlier studies [15,16].

In St. John’s wort extracts from agitated cultures maintained for 1–5 weeks, the presence of metabolites was found from the groups of phenolic acids, flavonoids, and catechins. The total content of the metabolite groups was calculated by summing up all individual compounds confirmed by the HPLC method. These were simple phenolic acids: protocatechuic acid, 3,4-dihydroxyphenylacetic acid, and vanillic acid; depsides: chlorogenic acid and its isomers, cryptochlorogenic acid and neochlorogenic acid; flavonoid aglycones: luteolin, kaempferol, and quercetin; flavonoid glycosides: hyperoside, rutoside, and quercitrin; catechins: catechin and epicatechin. Among them, quercetin was the quantitatively dominant metabolite in all extracts.

These compounds are characteristic of the studied *Hypericum* species [3,18]. Only in the case of phenolic acids, the spectrum of compounds confirmed in the extracts is wider than in the soil-grown plants. Commonly, St. John’s wort contains only caffeic and chlorogenic acids.

Total phenolic acid content ranged from 76 to 446 mg/100 g of DW for cv. Elixir, from 77 to 505 mg/100 g of DW for cv. Helos, and from 46 to 381 mg/100 g of DW for cv. Topas. The highest content was obtained in the fifth week of growth in all cultivars. In the case of the Elixir cultivar, the shapes of accumulation curves in different medium variants are similar. For all three cultivars, the MS medium variant containing 0.1 mg/L of BAP and NAA each was the most optimal among those used (Figure 1).

The total contents of flavonoids obtained ranged from 207 to 2348 mg/100 g of DW for cv. Elixir, from 127 to 2386 mg/100 g of DW for cv. Helos, and from 169 to 1568 mg/100 g of DW for cv. Topas. The highest content was obtained in the fifth week of growth as well. At all time points and for all cultivars, the variant of the medium containing 0.1/0.1 mg/L BAP and NAA was by far the most conducive to the accumulation of flavonoids (Figure 1).

Data on catechin accumulation showed that the maximum total catechin content was 712 mg/100 g DW for the Helos cultivar, 636 mg/100 g DW for the Elixir cultivar, and 496 mg/100 g DW for the Topas cultivar. For the Elixir and Helos cultivars, the maximum content of this group of metabolites was achieved in the fourth week of cultivation. For the Topas cultivar, it was the fifth week. The most diverse conclusions can be drawn after analyzing the selection of the medium variant. Only for cv. Topas, the MS variant containing the lowest concentration of growth regulators was the most favorable. For cv. Elixir, the variant containing 1.0 mg/L BAP and 1.0 mg/L NAA was the most favorable, while for cv. Helos, the results are not so clear. The two medium variants containing 2.0/2.0 and 1.0/1.0 mg/L of BAP and NAA can be considered the best for the accumulation of catechins in biomass (Figure 1).

Analyzing all the results obtained, it can be concluded that the Helos cultivar is the cultivar with the greatest biosynthetic potential and accumulates the largest amount of secondary metabolites from the groups of compounds studied.

Plants have developed a defense mechanism that includes the production of a wide variety of secondary metabolites which serve as tools to cope with environmental conditions. Even the physical conditions, medium composition, and time of growth can be factors in activating the production of species-specific secondary metabolites. Factors that affect the biosynthesis of hypericins, hyperforin, and flavonoids in wild and field-grown *H. perforatum* plants were described [19]. In the case of in vitro plant cell and organ cultures, it is assumed that, during the intensive growth phase, the biosynthesis of primary metabolites and building molecules takes place. After reaching a plateau, there is a switch to secondary metabolism. However, it is always necessary to confirm this fact experimentally and determine the optimal conditions for the biosynthesis and accumulation of individual metabolites [13,14].

Higher differentiated in vitro cultures, e.g., microshoot cultures, are capable of producing more secondary metabolites than undifferentiated cell cultures. Some research, connected with the biosynthesis and accumulation of flavonoids, documented that the richest source of the flavonoid fraction was shoot cultures. It was also documented for shoot cultures of *H. perforatum* [20]. The lower degree of organogenesis, e.g., callus cultures with shoot buds, callus cultures, and cell suspensions, decreased the diversity and overall flavonoid content [20,21]. *H perforatum* cv. Topas callus and shoot cultures were shown to produce hypericin and pseudohypericin, whereas hyperforins, chlorogenic acid, and flavonoids, such as quercetin, hyperoside, and rutoside, were only detected in shoot cultures [22]. The experimental culture model we have chosen is based not only on our previous experiments, but also on data from the literature. Savio et al. [23] investigated the in vitro micropropagation of St. John’s wort related to the secondary metabolism of different cultivation systems. They demonstrated that liquid systems of shoot cultivation (total immersion, partial immersion, paper bridge support) resulted in a higher phenolic content than the semi-solid system. The choice of type and quantity of PGRs represent the most significant factors in the regulation of morphogenesis and metabolism in vitro [24,25]. *H. perforatum* extracts from PGR-treated samples generally showed improved bioactivities compared to controls [26]. The effect of cytokinins was to increase the content of hypericin and hyperforin in shoot cultures of *H. perforatum*. This effect was demonstrated in in vitro cultures of different species of *Hypericum* [14]. Gadzovska [27] reported that low BAP concentrations (up to 1.0 mg/L) increased hypericin and pseudohypericin production in *H. perforatum* shoot culture, while higher BAP concentrations (1.0–2.0 mg/L) resulted in the inhibition of secondary metabolite production. Studies of *H. calycinum* showed that supplementation of culture medium with low concentrations of BAP (0.2 mg/L) and IBA (0.1 mg/L) slightly affected the level of polyphenolics, whereas indole-3-butyric acid (IBA) in higher concentrations inhibited polyphenolic levels in vitro [28].

### 2.2. Cultures in Bioreactors

To ensure that the agitated culture is the best experimental model for further experiments, cultures were also carried out in commercially available temporary immersion system bioreactors (PlantForm bioreactors), especially designed for shoot cultures.

The biomass of in vitro cultures of three *H. perforatum* cultivars grown in PlantForm bioreactors (MS medium supplemented with BAP and NAA, 4-week growth cycle) in the same MS medium variants as agitated cultures, did not significantly differ in morphology when MS medium variants with different contents of plant growth regulators were used (Figure 3). They were characterized by strongly leafy shoots, with a predominance of green parts over callus ones. The least vigorous cultures of all cultivars were observed at BAP/NAA concentrations of 3.0/3.0 mg/L. The increase in dry biomass in all cultivars was not very high, ranging from 4.35 to 5.27 times. The biomass of the Helos cultivar was the best growing in the bioreactors (Figure 4). Only slightly weaker increases were shown for the biomass of the Topas cultivar.

The presence of phenolic acids (protocatechuic, neochlorogenic, 3,4-dihydroxyphenylacetic, chlorogenic, cryptochlorogenic, and vanillic acids) was demonstrated in extracts from bioreactor cultures of the *H. perforatum* cultivars. They were the same as in the case of agitated cultures. The total content of this group of metabolites ranged from 210 to 353 mg/100 g of DW for cv. Elixir, from 163 to 336 mg/100 g of DW for cv. Helos, and from 220 to 287 mg/100 g of DW for cv. Topas (Figure 4). For all cultivars, the highest accumulation was found in biomass grown in MS medium containing 3 mg/L BAP and NAA each. This is the opposite result to the results obtained for agitated cultures. Compared to the maximum content in agitated cultures of 4 weeks, biomass in bioreactors accumulated a larger amount of phenolic acids: 1.44 times for the Elixir cultivar, a comparable amount for the Helos cultivar, and slightly less (1.16-fold) for the Topas cultivar.

The analysis of flavonoid compounds showed the presence of the same compounds in both types of cultures (hyperoside, rutoside, quercitrin, quercetin, luteolin, and kaempferol). The main compound was quercetin. The total content of flavonoids ranged from 609 to 881 mg/100 g of DW for the Elixir cultivar, 685 to 954 mg/100 g of DW for the Helos cultivar, and 705 to 1101 mg/100 g of DW for the Topas cultivar (Figure 4). As in the case of phenolic acids, all cultivars exhibited the highest accumulation of flavonoids in the biomass cultivated on the medium containing 3 mg/L BAP and NAA, each. Compared to the maximum content in agitated cultures maintained for the same period, the biomass in bioreactors accumulated a lower content of flavonoids for all cultivars, namely 1.75 times for cv. Elixir, 1.81 times for cv. Helos, and 1.21 times for cv. Topaz.

In the case of compounds in the group of catechins, the analysis showed the presence of catechin and epicatechin in biomass from bioreactor cultures, as well as in agitated cultures. The total catechin content varied depending on the cultivar of *H. perforatum* studied. For cv. Elixir, the content ranged from 234 to 295 mg/100 g DW and was the highest in the medium containing 3 mg of PGRs. For cv. Helos, the content ranged from 184 to 375 mg/100 g DW and was the highest in the biomass cultivated on the medium variant with 2.0/2.0 mg/L BAP and NAA. In cv. Topas, the content of catechins ranged from 138 to 344 mg/100 g of DW (Figure 4). For this cultivar, the maximum content of catechins was confirmed, in contrast to other cultivars and groups of metabolites in the biomass grown in the medium with the lowest concentration of BAP and NAA (0.1 mg/L each). When comparing these data with agitated cultures, the contents were 2.15, 1.90, and 1.28 times lower for the Elixir, Helos, and Topas cultivars, respectively. In the case of cultures carried out in bioreactors, the medium most conducive to the accumulation of metabolites was the variant containing 3 mg/L BAP and NAA, each.

Zobayed et al. (2003) described the growth of the culture and the content of metabolites (hypericin, pseudohypericin, hyperforin) in shoot cultures exposed to elevated levels of CO_2_ and/or sucrose in a balloon-type bubble bioreactor system (BB bioreactor) [29]. The same authors investigated St. John’s wort cv. ‘New Stem’ shoot culture systems, including a balloon-type bubble bioreactor, a temporary immersion bioreactor, and a temporary immersion bioreactor in the root zone. The balloon-type bioreactor was the most conducive to biomass production, but for hypericin, pseudohypericin, and hyperforin accumulation, stationary cultures in the gelled medium were much more suitable [30]. 

A more promising type of *H. perforatum* organ culture in bioreactors is adventitious root culture [31,32]. The culture was maintained in balloon-type airlift bioreactors to check the effect of inoculation density, air volume, IBA concentration, and methyl jasmonate elicitation on hypericin content [32]. Other researchers focused on large-scale 100 L, 500 L balloon-type and 500 L horizontal drum-type airlift bioreactors of *H. perforatum,* examining adventitious roots growth and metabolites production. They reported possibilities of the production of phenolics, flavonoids, chlorogenic acid, hyperin (hyperoside), hypericin, and quercetin in pilot-scale bioreactors [33]. The highest amount of hyperin and quercetin determined were 0.01 and 0.02 mg/g DW, respectively, which is much lower than what we obtained in shoot cultures (max. 153 and 712 mg/100 g DW, respectively). On the contrary, the chlorogenic acid content (1.3 mg/g DW) was 21 times higher. 

The optimization of cultivation depends on the type of culture, the type of bioreactor used, and its operation, e.g., aeration and immersion time, as well as standard cultivation conditions [34]. This has been demonstrated by us for both *Schisandra chinensis* [35] and *Verbena officinalis* cultures [36], where different types of bioreactors were tested. The accumulation of flavonoids and phenolic acids was also tested in shoot cultures of *Scutellaria baicalensis, S. lateriflora*, and *Schisandra chinensis* cv. Sadova. For both *Scutellaria* species, the total flavonoid content was 1.7 to 2.8 times higher in bioreactor culture biomass than in stationary or agitated cultures [37,38]. The contents of flavonoids and phenolic acids in *S. chinensis* cv. Sadova cultures grown in bioreactors were comparable to those confirmed in agar cultures [39]. 

The advantage of cultivation in bioreactors is the high increase in biomass, which in combination with biosynthesis and accumulation of valuable, biologically active metabolites, can be of practical use on a commercial level. Taking into account both the biomass increments and the content of secondary metabolites, we considered agitated cultures a better model for future research. For this purpose, we selected extracts with the highest secondary metabolite content of each *H. perforatum* cultivar to test their antioxidant potential, in correlation with the total phenolic, flavonoid, and condensed tannin content, as well as the antimicrobial activity. In addition, the *Artemia salina* lethality bioassay was performed with the aim of establishing the potential safety of the extracts.

### 2.3. Total Phenolic, Flavonoid, and Condensed Tannin Contents of Selected Extracts

The estimation of the phenolic content of *H. perforatum* cv. Elixir, cv. Helos, and cv. Topas biomass extracts (MS, 0.1 mg/L BAP and 0.1 mg/L NAA, 5 weeks) was undertaken using Folin–Ciocalteu phenol reagent. The amount of total phenolics of *H. perforatum* extracts ranged from 69 to 76 mg GAE/g extract, revealing that cv. Helos is the richest (75.99 ± 0.19 mg GAE/g extract). The polyphenol content of the extracts decreased in the order cv. Helos ≥ cv. Topas > cv. Elixir. These results are consistent with those described by Kwiecień et al. [15], who reported the highest content of phenolics in methanol extract obtained from the shoot culture of *H. perforatum* cv. Helos. A similar trend was observed for the condensed tannins; in fact, cv. Helos was found to have the highest level (19.85 ± 0.36 mg CE/g extract). The condensed tannins content, determined by spectrophotometric determination, decreased in the order cv. Helos ≥ cv. Topas > cv. Elixir.

On the contrary, the content of total flavonoids showed a different trend, ranging from 48.33 ± 0.27 mg QE/g extract (cv. Topas) to 23.21 ± 0.29 mg QE/g extract (cv. Helos) (Table 1). These results agree with those obtained by HPLC analysis, concluding that Helos is the cultivar with the greatest biosynthetic potential.

### 2.4. Activity of Selected Extracts

#### 2.4.1. Antioxidant Activity

The antioxidant properties of polyphenols are due to their redox properties, which allow them to act as reducing agents, hydrogen donators, metal chelators and single oxygen quenchers. Polyphenolics exhibit a wide range of biological effects, including antibacterial, anti-inflammatory, antiallergic, hepato-protective, antithrombotic, antiviral, anticarcinogenic, and vasodilatory actions; many of these biological functions have been attributed to their free radical scavenging and antioxidant activities [40]. In view of these potential health benefits, there has been intensive research on natural antioxidants derived from the plant kingdom. Antioxidants can be divided into primary (or chain-breaking) and secondary (or preventive) compounds, and the primary antioxidant reactions can be classified as hydrogen-atom transfer (HAT) and single-electron transfer (SET) reactions. The HAT mechanism takes place when an antioxidant scavenges free radicals donating hydrogen atoms; an antioxidant acting by the SET mechanism transfers a single electron to reduce any compound.

In order to broadly characterize the antioxidant potential of extracts, different methods should be utilized. Thus, the primary antioxidant activity of *H. perforatum* cv. Elixir, cv. Helos, and cv. Topas biomass extracts (MS, 0.1 mg/L BAP and 0.1 mg/L NAA, 5 weeks) was evaluated using the DPPH test (involving both HAT and SET mechanisms) and the reducing power assay (based on the SET mechanism); the secondary antioxidant ability was assessed by determining the ferrous ions’ chelating activity. 

The results of the DPPH assay showed that all extracts exhibited a radical scavenging effect that increased with increasing concentrations (Appendix A). Both cultivars Topas and Helos displayed good radical scavenging properties (IC_50_ = 0.512 ± 0.036 mg/mL and 0.586 ± 0.062, respectively) (Table 2). At the higher doses tested, the activity was close to that of the standard BHT (about 90%) (Appendix A). The extract obtained from the in vitro culture of cv. Elixir (IC_50_ = 1.625 ± 0.048 mg/mL) was much less active in the DPPH test. Based on IC_50_ values, the activity of the extracts and the standard decreases in the order BHT > *H. perforatum* cv. Topas ≥ *H. perforatum* cv. Helos > *H. perforatum* cv. Elixir (Table 2). The linear regression analysis revealed a strong positive correlation between DPPH radical activity and both polyphenols and condensed tannins (R^2^ = 0.9848 and R^2^ = 0.9056, respectively). No correlation with flavonoids was found. 

As shown in Appendix A, *H. perforatum* extracts showed mild reducing power. The effect of the three extracts was dose-dependent and resulted in lower values than those of BHT utilized as a reference drug. Based on ASE values, cv. Elixir manifested the best activity followed by cv. Helos and cv. Topas (Table 2). A positive linear correlation between reducing power and both polyphenols and flavonoids was found (R^2^ = 0.634 and R^2^ = 0.7072, respectively). A mild correlation with condensed tannins is highlighted (R^2^ = 0.4472).

In the Fe^2+^ chelating activity assay, all the extracts exhibited good, dose-dependent, chelating properties, reaching about 80% at the highest concentration tested (Appendix A). Based on IC_50_ values, the activity of all the cultivars is superimposable, and cv. Helos was the most active one followed by cv. Topas and cv. Elixir (Table 2).

A strong linear correlation between chelating activity and polyphenols and a good correlation between chelating activity and condensed tannins was found (R^2^ = 0.8203 and R^2^ = 0.7947, respectively), whereas no correlation with flavonoids was detected. 

Based on the results obtained, it is evident that extracts from in vitro microshoot cultures of *H. perforatum* act as moderate primary antioxidants and have good secondary antioxidant properties in the Fe^2+^ chelating activity assay. The extracts contain large amounts of flavonoids, catechins, and phenolic acids. Flavonoids and phenolic acids are the largest classes of plant phenolics; several compounds in these classes have been shown to possess powerful antioxidant activity in both in vitro and in vivo investigations. Good examples are quercetin, myricetin, luteolin (from flavonoid group), caffeic acid, protocatechuic acid, and depsides: chlorogenic acid and its isomers (from phenolic acid group) [41,42]. Quercetin and its glycosides are the main flavonoids of *H. perforatum* and the tested biomass extracts. The antioxidant activity of quercetin has been extensively studied, including its effects on glutathione, enzymatic activity, signal transduction pathways, and reactive oxygen species (ROS). Chemical studies on quercetin have focused mainly on the antioxidant activity of its metal ion complexes with vanadium, copper, magnesium, iron, ruthenium, cobalt, cadmium, calcium, and rare earth elements, based on the DPPH free radical scavenging test [43]. Furthermore, phenolic acids, such as protocatechuic or chlorogenic acid and its isomers, exhibit antioxidant properties [44,45]. Silva et al. [46] reported on the antioxidant potential of the ethanolic extract of *H. perforatum*. They characterized the composition of the extract and connected free radical scavenging with the presence of quercetin and its glycosides as well as kaempferol and chlorogenic acid. The antioxidant activity of fractionated *H. perforatum* extract demonstrated that it is mainly attributed to flavonoid glycosides and phenolic acids (chlorogenic acid), while phloroglucinols, biflavonoids, and naphthodianthrones showed no significant activity [47].

Moreover, tannins have shown the ability to quench free radicals and their effectiveness depends on the high molecular weight, the number of aromatic rings, the nature of hydroxyl group substitution, and the specific functional groups [48].

With regard to secondary antioxidant properties, a good correlation with the content of condensed tannins was found, as indicated by the coefficient of determination. Antioxidant properties of tannins can result from their ability to chelate transition metal ions, especially Fe(II) and Cu(II). Metal ions can generate highly reactive oxygen free radicals by Fenton or Haber–Weiss chemistry. In the Fenton reaction, the hydroxyl radical (HO^•^) is produced from hydrogen peroxide. In the iron-catalyzed Haber–Weiss reaction, the superoxide anion radical (O_2_^•−^) reduces ferric to ferrous ions, which, in turn, are involved in generating hydroxyl radicals [49]. Extremely reactive hydroxyl radicals can interact with many biological macro- and small molecules and therefore lead to lipid peroxidation, DNA damage, and polymerization or denaturation of proteins. The binding of transition metal ions by tannins can stabilize the prooxidative activity of those ions [50,51]. Bibliographic data demonstrate that phenolic acids are strong metal chelators, too [52]. Thus, *H. perforatum* extracts obtained from in vitro culture of the three cultivars could offer protection against oxidative damage through their chelating properties correlated to the high content of phenolic acids and condensed tannins present in the extracts.

#### 2.4.2. *Artemia salina* Lethality Bioassay

*A. salina*, the brine shrimp, is an invertebrate that is used for the preliminary evaluation of the toxicity of bioactive compounds and plant extracts. The main advantages of using brine shrimp in toxicity testing are rapidity, cost-effectiveness, and simplicity. This assay may be considered an alternative to the in vitro cell culture [53] and in vivo assays. A good correlation between the results of the oral acute toxicity determination in the murine model and this bioassay has been previously reported; thus, it represents a useful tool for predicting the acute toxicity of plant extracts [54]. Aimed at establishing the potential safety of the extracts, an *A. salina* lethality bioassay was performed. All the tested extracts were found to be non-toxic against brine shrimps, (LC_50_ > 1000 μg/mL). The literature data demonstrated that *H. perforatum* extracts, which are particularly rich in hypericin, show a cytotoxic effect on cancer cell lines such as metastatic melanoma cells, HT-29 colon cancer cells, and HeLa cells. Hypericin itself was cytotoxic on cancer cells, e.g., MCF-7 human breast cancer [55]. However, when tested in vivo in the rat model, *H. perforatum* had no cytotoxic potential in the test system evaluated [56]. The potential safety of our extracts from in vitro cultures of St. John’s wort, as demonstrated by the *A. salina* lethality test, is probably due to the low content of hypericin [15].

#### 2.4.3. Antibacterial Activity

The studies showed (Table 3) that the extracts had a 3–4-times stronger effect on Gram-positive bacteria (MIC = 2.5–15.0 mg/mL) than on Gram-negative bacteria (MIC = 10.0–40.0 mg/mL). The strongest antibacterial activity was shown by an extract from in vitro cultures of *H. perforatum* cv. Helos, in particular against *Staphylococcus aureus* (MIC = 7.5 mg/mL), *Staphylococcus epidermidis* (MIC = 2.5 mg/mL), *Bacillus subtilis* (MIC = 2.5 mg/mL), and *Enterobacter cloacae* (MIC = 10 mg/mL). The extract from the Topas cultivar exhibited good activity against *Staphylococcus epidermidis* (MIC = 5.0 mg/mL) and *Bacillus subtilis* (MIC = 2.5 mg/mL). The weakest antibacterial activity among the tested extracts was confirmed for *H. perforatum* cv. Elixir.

Several studies are available in the literature on the in vitro antibacterial activity of crude plant extracts from aerial parts of *H. perforatum*, supporting the use of this plant in traditional medicine to treat wounds and skin diseases. The antimicrobial activity of St. John’s wort is primarily associated with the presence of hyperforin and hypericin [8]. Several studies have also confirmed the antibacterial effect of the essential oil of *H. perforatum* [57,58,59]. Hyperforin is the main compound involved in the antimicrobial activity against *Staphylococcus aureus*, multidrug-resistant *S. aureus*, and other Gram-positive bacteria, such as *Staphylococcus saprophyticus, S. epidermidis, Streptococcus pyogenes, Enterococcus* sp., *Listeria monocytogenes*, and *Corynebacterium diphtheriae* [60,61,62,63]. On the other hand, a good growth inhibitory effect has never been confirmed against Gram-negative bacteria, such as *Escherichia coli, Enterococcus faecalis*, and *Pseudomonas aeruginosa* [64,65]. Despite this, in our test, some activity of *H. perforatum* in vitro culture extracts against Gram-negative bacteria strains was observed. The extraction solvent selection and the obtained extract’s chemical composition play a significant role in its antibacterial activity. It was shown that the ethyl acetate extract, containing mainly flavonoids, hypericins, and hyperforins, was the most active against the Gram-positive bacteria tested. The inhibition of bacterial growth was correlated with the content of hypericin and hyperforin, whereas flavonoids appeared to be completely inactive [66]. The methanolic extracts tested by us from the in vitro culture of *H. perforatum* contain mainly phenolic compounds. In our previous article, we determined the hypericin content in the range of 3–14.5 mg/100 g of DW [15]. Therefore, the antibacterial activity confirmed by us against five strains of Gram-positive bacteria is a significant result. 

Mazandarani et al. [65] reported the high antibacterial activity of ethanolic extracts of *H. perforatum* herb against Gram-positive bacteria (*Enterococcus faecalis* and *Staphylococcus aureus*), with growth-inhibition zones in the range of 25–26 mm. These authors showed that alcoholic extract is most active in comparison with aqueous extracts and infusion. The methanolic extract from the dried aerial part of *H. perforatum* in broth culture was active on *Staphylococcus oxford* (MIC 0.62 mg/mL), and on *Staphylococcus aureus* (MIC 1.25 mg/mL). On agar plates with bacterial culture methanolic extract of the dried aerial part (20.0 µL/disc), it was active on *Escherichia coli*, weakly active on *Bacillus subtilis*, hardly active on *Pseudomonas aeruginosa* and *Staphylococcus aureus*, and inactive on *Enterobacter aerogenes*, *Klebsiella pneumoniae, Salmonella typhimurium,* and *Serratia marcescens* [67]. For this reason, the demonstration in our study of moderate bacteriostatic activity against five strains of Gram-negative bacteria deserves to be emphasized.

#### 2.4.4. Antifungal Activity

The activity of the extracts analyzed against yeast-like strains of the genus *Candida* was slightly stronger (MIC = 7.5–30 mg/mL) compared to its activity against mold fungi (MIC = 15.0–40.0 mg/mL). The strongest fungistatic activity was shown by an extract from in vitro cultures of the Elixir cultivar, especially against *Candida albicans* and *Candida krusei* (MIC = 7.5 mg/mL). On the other hand, extracts of all the analyzed *Hypericum* cultivars were characterized by the highest activity against the dermatophyte *Trichophyton tonsurans* (MIC = 2.5–7.5 mg/mL) (Table 4).

Information on the antifungal activity of St. John’s wort is limited. One of the studies demonstrated the diverse efficacy of hypericin against *Candida albicans, Exophiala dermatitidis, Microsporum canis, Fusarium oxysporum, Trichophyton rubrum, Pichia fermentans, Kluyveromyces marxianus*, and *Saccharomyces cerevisiae* [68]. In contrast, methanolic extract of the dried aerial part, on an agar plate at a concentration of 80.0 mg/disc, was inactive on *Aspergillus flavus, Aspergillus fumigatus, Fusarium tricinctum, Trichoderma viride,* and *Trichophyton mentagrophytes*. It showed weak activity on *Microsporum cookei* and *Microsporum gypseum* [1]. Milosevic et al. [69] showed the antifungal activity of an ethanolic extract of *H. perforatum* against the fungi *Fusarium oxysporum* and *Penicillium canescens* at a concentration of 45 mg/mL. Inhibition of the growth of the same fungal species by the ethanolic extract of *H. perforatum* using the dilution assay (MIC = 10 mg/mL) and microscopic methods was confirmed by other authors [70]. Our study demonstrates the diverse antifungal activity of methanolic extracts from in vitro cultures. It is worth noting the activity of all biomass extracts against *Trichophyton tonsurans* and the activity of the cv. Elixir extract on *Candida* sp.

### 2.5. Enhancing the Production of Secondary Metabolites by Phenylalanine Feeding

The culture medium was supplemented with phenylalanine, a biogenetic precursor of many plant metabolites with a phenolic structure. Agitated cultures were grown for 4 weeks in MS medium containing 0.1 mg/L BAP and 0.1 mg/L NAA. After this time, the precursor was administered. The content of metabolites was determined, 2, 4, and 7 days after the addition of phenylalanine. The conditions of the experiment applied, i.e., the concentration of phenylalanine and the culture time that elapsed from the supplementation of the medium with the precursor, did not negatively affect the biomass growth of the agitated cultures of all three tested cultivars of St. John’s wort. Phenylalanine supplementation influenced the appearance of biomass. It had a slightly darker green color and a more compact structure compared to the control samples.

Qualitative analysis allowed to the confirmation of the presence of the previously determined metabolites in all extracts tested, i.e., protocatechuic acid, 3,4-dihydroxyphenylacetic acid, chlorogenic acid, neochlorogenic acid, cryptochlorogenic acid, quercetin, luteolin, kaempferol, hyperoside, rutoside, quercitrin, catechin, and epicatechin. Additionally, the presence of *p*-hydroxybenzoic acid and *p*-coumaric acid was confirmed in extracts from cultures supplemented with phenylalanine. The presence of vanillic acid was confirmed in selected extracts in trace amounts. The presence of apigetrin, epigallocatechin, and epicatechin gallate was also confirmed in biomass extracts after the administration of the precursor. The phytochemical profile obtained for the phenolic metabolites (phenolic acids, flavonoids, and catechins) in the control samples coincided with the results obtained in previous experiments.

The total phenolic acid content reached in the biomass of the Elixir cultivar after supplementation with phenylalanine was 388 mg/100 g of DW on the second day, 579 mg on the fourth day, and 771 mg on the seventh day. For cv. Helos, the total phenolic acid content increased from 371 mg/100 g DW on day two, to 590 mg on day four, and finally to 662 mg on day seven. While for cv. Topas, the phenolic acid content increased from 454 mg/100 g of DW on day two, to 536 mg on day four, and finally to 613 mg on day seven. All the above values were higher than the content of the control. For all cultivars, the highest content was confirmed on day 7 after the addition of phenylalanine. Compared to the control, the content of phenolic acids increased 1.73, 1.31, and 1.61 times, in the Elixir, Helos, and Topas cultivars, respectively. The main metabolites in this group were 3,4-dihydroxyphenylacetic acid, *p*-hydroxybenzoic acid, and neochlorogenic acid. The maximum contents of these compounds were 265 mg/100 g of DW for 3,4-dihydroxyphenylacetic acid (Elixir, day 7), 165 mg/100 g of DW for *p*-hydroxybenzoic acid (Helos, day 7), and 162 mg/100 g of DW for neochlorogenic acid (Helos, control sample, day 7) (Table 5). 

The total content of flavonoid compounds reached in the biomass of the Elixir cultivar after supplementation with phenylalanine was 1636 mg/100 g DW (day two), 2157 mg (day four), and 3086 mg (day seven). For the Helos cultivar, the total content of flavonoids increased from 1828 mg/100 g DW (day two), to 1961 mg (day four), and finally to 2887 mg (day seven), whereas for cv. Topas, the flavonoid content reached 2013 mg/100 g DW (day two), 2158 mg (day four), and 3345 mg (day seven). All the mentioned values were higher than the content in the control samples. For all cultivars, the highest content was confirmed on day 7 after the addition of phenylalanine. Compared to the control, this content increased 1.25, 1.21, and 2.23 times, in the Elixir, Helos, and Topas cultivars, respectively. The dominant metabolite in this group was quercetin. The maximum content of this compound was 2797 mg/100 g of DW (Topas, day 7) (Table 5). Compared to the control, the content of quercetin increased 1.51, 1.46, and 2.09 times, in the Elixir, Helos, and Topas cultivars, respectively.

The total catechin content in the biomass of the Elixir cultivar after supplementation with phenylalanine was 539 mg/100 g of DW on day two. It reached the maximal value of 717 mg on day four, and decreased thereafter to 622 mg on day seven. For the Helos cultivar, the total catechin content increased from 552 mg/100 g of DW on day two, to the maximal content of 660 mg on day four, and decreased to 573 mg on day seven. For cv. Topas, the catechin content was the lowest, namely, 454 mg/100 g DW on day two, 424 mg on day four, and 173 mg on day seven. For all cultivars, the highest content of catechins was confirmed on day 4 after the addition of phenylalanine. The maximum total content of catechins for the Elixir cultivar was 1.59 times higher than the control result. The maximum total content for the Helos cultivar was 1.38 times higher than the control. However, the maximum total content for the Topas cultivar after administration was lower and reached 96% of the catechin content in the control sample. The main metabolites in this group were catechin and epicatechin. The maximum content of these compounds was 312 mg/100 g DW catechin (Helos, day 7, control sample), and 461 mg/100 g DW epicatechin (Elixir, day 7) (Table 5).

Under the conditions of the experiment, Elixir was the cultivar that accumulates the highest phenolic acids and catechins levels, while the highest content of flavonoids was confirmed in the Topas cultivar. When summarizing the results, *H. perforatum* cultivar with the highest biosynthesis potential and accumulation capacity of the determined metabolites after supplementation with phenylalanine is cv. Elixir (4.48 g of estimated polyphenols in 100 g of DW). The results obtained for the other two cultivars Helos and Topas are slightly lower but also relatively high, namely, 4.12 and 4.13 g/100 g DW, respectively.

Only a few papers reported precursor feeding in in vitro shoot cultures of *H. perforatum*. In liquid cultures, the feeding of L-phenylalanine, L-tryptophan, cinnamic acid, and emodin was tested. The effects of these precursors on the production of hyperforin, pseudohypericin, and hypericin differed. The addition of phenylalanine enhanced the production of hypericins but decreased the levels of hyperforins. All metabolites decreased when tryptophan was added to the medium, while cinnamic acid and emodin each enhanced the accumulation of hyperforin in *H. perforatum*, but did not affect the level of hypericins [71]. Other research showed that in liquid shoot cultures feeding with L-threonine and L-isoleucine enhanced adhyperforin production rather than hyperforin production [72]. Emodin increased the production of pseudohypericin and hypericin, administered alone and with sodium acetate. A moderate effect on the stimulation of hypericin production by succinic and malic acid was reported [14]. So far, all feeding experiments are related only to hypericins and hyperforin contents. 

Phenylalanine is quite often administered in in vitro cultures as a precursor of phenolic compounds. The phenolic secondary metabolites in plants are derived from phenylalanine, itself being formed via shikimate biosynthesis. The intermediate compound, *p*-coumaroyl-CoA, forms the origin of various metabolites. In shoot cultures of a few species, the influence of phenylalanine administration on the biosynthesis of phenolic metabolites was investigated. An increase in the content of the following metabolites has been demonstrated: a 1.6-fold increase of phenolic acids in *Exacum affine* [73]; 1.8- and 2.8-fold increases of total phenolic acids in *Aronia melanocarpa* and *A. arbutifolia*, respectively [74]; a 2.4-fold increase of total polyphenols in *Nasturtium officinale* [75]; a 3.9-fold increase of verbascoside in *Scutellaria baicalensis* [37]; and a 2.2-fold increase of total flavonoids in *S. lateriflora* [38]. It has been shown that the optimum phenylalanine concentration that can stimulate the production of phenolic acids in shoot cultures varies widely from 0.016 g/L for *A. melanocarpa* to 1.6 g/L for *E. affine*. Less differentiated cultures, like suspension or callus cultures, also utilize phenylalanine to biosynthesize phenolic metabolites, such as flavonoids in the callus of *Abutilon indicum* [76], anthocyanidins in the suspension culture of *Panax sikkimensis* [77], or phenolic acids in the callus culture of *Ginkgo biloba* [41]. 

As in the case of the optimization of the culture process, the optimal concentration of a supplemented precursor should be selected experimentally for each plant species and the type of the tested culture. Phenylalanine is usually used as a substrate for metabolite biosynthesis and a high concentration is needed to obtain a satisfactory result. This amino acid can act as an enzymatic activator of phenylalanine ammonia-lyase, and even a low concentration of phenylalanine is often sufficient to stimulate the metabolism.

## 3. Materials and Methods

### 3.1. In Vitro Initial Cultures

The in vitro cultures of three *H. perforatum* L. cultivars, Elixir, Helos, and Topas, were established in 2007 at the Institute of Pharmaceutical Biology, Technische Universität Braunschweig (Germany) [17]. The cultures were grown in Erlenmeyer flasks (250 mL) on Murashige and Skoog (MS) [78] medium, solidified with agar (Phyto agar, Duchefa Biochemie, Haarlem, The Netherlands) and supplemented with 0.5 mg/L of BAP and NAA each, under constant LED light with an intensity of 16 μmol/m^2^ s at 25 ± 2 °C. They were subcultured every 6 weeks. Stationary agar cultures were used as a material to establish experimental cultures: agitated and bioreactor cultures.

### 3.2. Agitated Cultures

The agitated cultures were initiated with an inoculum of 1 g per flask of fresh biomass with one to three clusters of shoots. Agitated shoot cultures of the three cultivars of *H. perforatum* were maintained in Erlenmeyer flasks in four different variants of MS medium, which contained 0.1, 1.0, 2.0, or 3.0 mg/L of BAP and NAA each, as previously described by Kwiecień et al. [15]. Each flask (500 mL) contained 100 mL of medium. Cultures (three replicate flasks) were maintained under the same light and temperature conditions as the initial cultures, with agitation carried out using a rotary shaker (Altel, Łódź, Poland) operating at 140 rpm with an amplitude of 35 mm for 1 to 5 weeks.

### 3.3. Cultures in Bioreactors

The biosynthetic potential of microshoot cultures of the *H. perforatum* cultivars was also tested in the bioreactor culture model. The commercially available temporary immersion system (TIS) PlantForm bioreactors (PlantForm, Sweden), designed for microshoot culture, were inoculated with 15 g of biomass per 500 mL of medium supplemented with the same sets of PGRs as agitated cultures (three replicates each). The immersion frequency was 5 min every 90 min. Cultures were grown for 4 weeks under light and temperature conditions as described above for agar and agitated cultures.

### 3.4. Feeding Culture Medium with Phenylalanine as the Biogenetic Precursor

An MS medium supplemented with 0.1 mg/L BAP and 0.1 mg/L NAA was chosen for the agitated cultures as the best productive medium for further experiments. The cultures were maintained for 4 weeks on a rotary shaker. The other cultivation conditions were the same as those mentioned for the agar, agitated, and bioreactor cultures. After 4 weeks, a biosynthetic precursor of phenolic compounds (phenylalanine) was added to the medium at the final concentration of 1.0 g/L. The dose of the precursor was chosen based on our previous study with *Scutellaria* species [33,34]. All experimental and control cultures (three replicate flasks each) were grown for an additional 2, 4, or 7 days.

### 3.5. Extracts Preparation

After each experiment, the fresh biomass was collected and dried at room temperature. The biomass increment was calculated by dividing the dry weight of the sample (DW) by the dry weight of the inoculum. The dried plant material was weighed, pulverized, and extracted by methanol (50 mL, 3 h, at 78 °C). After filtration, the methanol was evaporated. The dry residues were stored in the refrigerator at +4 °C until analysis was performed.

### 3.6. Reverse-Phase High-Performance Liquid Chromatography (RP-HPLC) Analysis

The samples were analyzed using a modified HPLC method [79]. A HPLC system (Merck–Hitachi) and a Purospher RP-18e analytical column (4 × 250 mm, 5 μm; Merck) were used. The mobile phase consisted of methanol (A) and 0.5 % acetic acid (B) (gradient elution). The flow rate was 1 mL/min. Compounds were estimated using a DAD detector. Qualification and quantification analyses were based on a comparison with 68 reference substances. Phenolic acids: 3,4-dihydroxyphenylacetic acid, 3-hydroxyphenylacetic acid, caftaric acid, caffeic acid, chlorogenic acid, cryptochlorogenic acid, 2-coumaric, 3-coumaric, 4-coumaric acids, dihydrocaffeic acid, ellagic acid, ferulic acid, 4-*O*-feruloyl-quinic acid, gallic acid, gentisic acid, hydrocaffeic acid, 4-hydroxybenzoic acid, isochlorogenic acid, isoferulic acid, neochlorogenic acid, protocatechuic acid, rosmarinic acid, salicylic acid, sinapic acid, syringic acid, and vanillic acid (Sigma-Aldrich®, St Louis, MO, USA). Flavonoids: apigenin, chrysin, cynaroside, isorhamnetin, hyperoside, luteolin, narigenin, myricetin, populnin, quercetin, quercitrin, rhamnetin, robinin, rutoside, and vitexin (Sigma-Aldrich®, St Louis, MO, USA); apigenin 5-glucoside, apigenin 7-glucuronide, apigenin 4′-rhamnoside, astragalin, avicularin, diosmetin, isoquercetin, kaempferol, kaempferol 3-glucoside, kaempferol 3-rhamnoside, kaempferol 7-rhamnoside, kaempferol 4-glucoside, miquelianin, narirutin, and quercetin 3-glucuronide (ChromaDex, Irvine, CA, USA); apigetrin, isovitexin, gujaverin, oroxylin A, trifolin, and vicenin II (ChemFaces, Wuhan, PRC). Catechins: catechin, epicatechin, epigallocatechin, epigallocatechin gallate, and epicatechin gallate (ChromaDex, Irvine, CA, USA).

### 3.7. Total Phenolic, Flavonoid, and Condensed Tannin Contents

The total phenolic content of *H. perforatum* cv. Elixir, cv. Helos, and cv. Topas methanol biomass extracts (MS, 0.1 mg/L BAP and 0.1 mg/L NAA, 5 weeks) were measured using the Folin–Ciocalteu reagent [80]. The solution (100 μL) containing a suitable concentration of each MeOH extract was mixed with 200 μL Folin–Ciocalteu reagent, 2 mL of distilled water, 1 mL of 15 % sodium carbonate, and incubated at room temperature in the dark for 2 h. The absorbance was then measured with a spectrophotometer (UV-1601 spectrophotometer, Shimadzu, Kyoto, Japan) at 765 nm. Gallic acid was used as a standard. Total phenolics were expressed as mg gallic acid equivalents (GAE)/g extract (DW) ± standard deviation (SD). 

The total flavonoid content of the extracts was measured using the aluminum chloride colorimetric assay [81]. A 500 μL aliquot of each sample solution appropriately diluted was mixed with 1.5 mL MeOH, 100 μL of 10 % aluminum chloride, 100 μL of 1 M potassium acetate, and 2.8 mL of distilled water. The samples were incubated at room temperature in the dark for 30 min, and the absorbance of the reaction mixture was measured at 415 nm. Quercetin standard was used to make the calibration curve and total flavonoids were expressed as mg equivalents of quercetin (QE)/g of extract (DW) ± SD. 

The condensed tannin content of the extracts was determined using the vanillin method, as previously described [82]. Each sample solution (50 μL) was mixed with 1.5 mL of 4 % vanillin in MeOH and 750 μL of concentrated hydrochloric acid. After incubation at room temperature in the dark for 20 min, the absorbance of the reaction mixture was measured at 500 nm. (+)-Catechin was used as the standard compound to make the calibration curve and condensed tannins were expressed as mg catechin equivalents (CE)/g of extract (DW) ± SD. The results of the spectrophotometric determinations were obtained from the average of three independent experiments.

### 3.8. Antioxidant Activity

Antioxidant activity occurs through different mechanisms; thus, no single testing method is capable of providing a comprehensive view of the antioxidant profile of a sample. Therefore, to evaluate the antioxidant capacity of plant-derived phytocomplexes or isolated compounds, various methods based on different approaches and mechanisms must be used. To determine the in vitro antioxidant effectiveness of methanol biomass extracts (MS, 0.1 mg/L BAP and 0.1 mg/L NAA, 5 weeks) from the *H. perforatum* cultivars Elixir, Helos, and Topas, three in vitro tests based on different mechanisms of determination of antioxidant capacity were used: DPPH (1,1-diphenyl-2-picrylhydrazyl), reducing power, and ferrous ions chelating activity.

#### 3.8.1. Free Radical Scavenging Activity

The free radical scavenging activity of the biomass extracts of *H. perforatum* was determined using the DPPH (1,1-diphenyl-2-picrylhydrazyl) method [83]. Extracts were tested at different concentrations (0.0625–1.75 mg/mL). An aliquot (0.5 mL) of a methanolic solution containing different amounts of sample solution was added to 3 mL of daily prepared methanolic DPPH solution (0.1 mM). The optical density change at 517 nm was measured, 20 min after initial mixing using a model UV-1601 spectrophotometer (Shimadzu). Butylated hydroxytoluene (BHT) was used as a reference. 

The scavenging activity was measured as the decrease in the absorbance of the samples versus the DPPH standard solution. The results were expressed as the percentage of radical scavenging activity (%) of the DPPH, defined by the formula [(Ao − Ac)/Ao] × 100, where Ao is the absorbance of the control and Ac is the absorbance in the presence of the sample or standard.

The results, obtained from the average of three independent experiments, are reported as mean radical scavenging activity percentage (%) ± standard deviation (SD) and mean 50% inhibitory concentration (IC_50_) ± SD.

#### 3.8.2. Reducing Power Assay

The reducing power of *H. perforatum* in vitro extracts was evaluated by spectrophotometric detection of the Fe^3+^-Fe^2+^ transformation method [84]. The extracts were tested at different concentrations (0.0625–1.75 mg/mL). Different amounts of samples in 1 mL solvent were mixed with 2.5 mL of phosphate buffer (0.2 M, pH 6.6) and 2.5 mL of 1% potassium ferricyanide [K_3_Fe(CN)_6_]. The mixture was incubated at 50 °C for 20 min. The resulting solution was rapidly cooled, mixed with 2.5 mL of 10% trichloroacetic acid, and centrifuged at 3000 rpm for 10 min. The resulting supernatant (2.5 mL) was mixed with 2.5 mL of distilled water and 0.5 mL of 0.1% fresh ferric chloride (FeCl_3_), and the absorbance was measured at 700 nm after 10 min; the increased absorbance of the reaction mixture indicates an increase in reducing power. As a blank, an equal volume (1 mL) of water was mixed with a solution prepared as described above. Ascorbic acid and BHT were used as references. The results, obtained from the average of three independent experiments, are expressed as mean absorbance values ± SD. The reducing power was also expressed as an ascorbic acid equivalent (ASE/mL).

#### 3.8.3. Ferrous Ions (Fe^2+^) Chelating Activity

The Fe^2+^ chelating activity of *H. perforatum* extracts was estimated by measuring the formation of the Fe^2+^-ferrozine complex, according to the method previously reported [85]. The extracts were tested at different concentrations (0.0625–1.75 mg/mL). Briefly, different concentrations of each sample in 1 mL solvent were mixed with 0.5 mL of methanol and 0.05 mL of 2 mM FeCl_2_. The reaction was initiated by adding 0.1 mL of 5 mM ferrozine. Then, the mixture was shaken vigorously and left standing at room temperature for 10 min. The absorbance of the solution was measured spectrophotometrically at 562 nm. The control contains FeCl_2_ and ferrozine, complex formation molecules. Ethylenediaminetetraacetic acid (EDTA) was used as a reference. The percentage of inhibition of the formation of the ferrozine-(Fe^2+^) complex was calculated by the formula [(Ao − Ac)/Ao] × 100, where Ao is the absorbance of the control and Ac is the absorbance in the presence of the sample or standard. The results, obtained from the average of three independent experiments, are reported as mean inhibition of the formation of the ferrozine complex (Fe^2+^) (%) ± SD and IC_50_ ± SD.

### 3.9. Artemia Salina Lethality Bioassay

In order to investigate the potential toxicity of *H. perforatum* in vitro extracts, the median lethal concentration (LC_50_) was determined according to a method previously reported [86]. The extracts, opportunely dissolved and then diluted in artificial seawater, were tested at final concentrations of 10, 100, 500, and 1000 µg/mL. Ten brine shrimp larvae were transferred to each sample vial and artificial seawater was added to obtain a final volume of 5 mL. After 24 h of incubation (25–28 °C), the surviving larvae were counted. The assay was carried out in triplicate and the LC_50_ values were determined by the Litchfield and Wilcoxon method. Extracts are considered non-toxic if the LC_50_ is higher than 1000 µg/mL.

### 3.10. Antimicrobial Activity

The tests were performed on sixteen standard and hospital strains of microorganisms. Among them, ten strains of Gram-positive and Gram-negative bacteria and six strains of fungi (yeasts, molds, and dermatophytes) were used. The list of selected strains is given in Table 6. The tested extracts were dissolved in DMSO at a concentration of 500 mg/mL, from which further dilutions were prepared in CASO Agar, Merck (Darmstadt, Germany) for bacteria, and Sabouraud Agar, Merck, for fungi in concentrations ranging from 1.0 to 50.0 mg/mL.

#### 3.10.1. Antibacterial Activity

The study was carried out using the plate agar dilution method. The following concentrations of the extracts: 50.0; 40.0; 30.0; 20.0; 15.0; 10.0; 5.0; 2.5 and 1.0 mg/mL were performed. Twenty-four-hour bacteria cultures were diluted in CASO broth (Merck). The inoculum, which contained 10^5^ CFU/mL was applied using a calibrated loop on the surface of the medium with the appropriate concentration of the tested extracts or without extracts (strain growth control). Incubation was carried out at 37 °C for 24 h. The MIC (Minimum Inhibitory Concentration) was determined as the lowest concentration of extract inhibiting the growth of bacteria on the agar. The reference substance was chloramphenicol (Merck, Darmstadt, Germany), for which the MIC value against the standard strain *Staphylococcus aureus* ATCC 6538 P was 0.005 mg/mL and against the Gram-negative strain *Escherichia coli* ATCC 8739 it was 0.1 mg/mL. 

#### 3.10.2. Antifungal Activity

The same procedure as for bacteria was followed for yeast-like fungi. For the tests, the plate dilution method on Sabouraud agar was used. The concentrations of the extracts were: 50.0; 40.0; 30.0; 20.0; 15.0; 10.0; 5.0; 2.5 and 1.0 mg/mL. Twenty-four-hour yeast cultures were diluted in Sabouraud broth (Merck) to obtain the cell density of 10^5^ CFU/mL. In turn, the cultured molds and dermatophytes were washed with fresh agar slants of these strains and then diluted in the same medium to obtain a density of 10^5^ CFU/mL. Then, the cultures on the surface of the agar plates with appropriate concentration of the extracts or without extracts (strain growth control) were inoculated with the tested strains of fungi using a calibrated loop. Incubation was carried out at 37 °C for 24 h (yeasts) and 5 days of incubation at 25 °C (molds and dermatophytes). The MIC was determined as the lowest concentration of extract inhibiting the growth of fungal strains on the agar. For amphotericin B, a reference substance (Serva, Heidelberg, Germany), the MIC against the standard strain *Candida albicans* PCM 1409 PZH was 1.0 mg/mL.

### 3.11. Statistical Analysis

Comparison of data obtained from spectrophotometric determinations and antioxidant tests was made using one-way ANOVA, followed by the Tukey–Kramer multiple comparison test (GraphPAD Prism Software for Science); *p*-values less than 0.05 were considered statistically significant. Other statistical analyses were performed using STATISTICA 13.3 software (TIBCO Software Co., Palo Alto, CA, USA). Differences in biomass growth and metabolite content in biomass were analyzed by ANOVA, followed by a Bonferroni post hoc test and *p*-values < 0.05. All experiments were performed with at least three independent replications. The results were expressed as means ± SD of the mean.

## 4. Conclusions

High biomass increments (max. 12.1-fold) and very high amounts of phenolic acids, flavonoids, and catechins in the biomass extracts from three cultivars (max. 505, 2386 and 712 mg/100 g DW, respectively) were obtained. This was achieved by performing extensive optimization of in vitro culture conditions of three *Hypericum perforatum* cultivars (Elixir, Helos, Topas), testing two different types of in vitro cultures (agitated and bioreactor cultures), studying different concentrations of PGRs (BAP and NAA) in MS culture medium, testing the duration of growth cycles, and seeking the best conditions for an increase in biomass and an accumulation of bioactive metabolites. The high contents of these antioxidants resulted in the documented interesting biological properties of the biomass extracts, namely antioxidant and antimicrobial activities. 

At this stage of the experiment, we propose the biomass of *H. perforatum* cultivars from an agitated culture maintained on MS medium with 0.1 mg/L BAP and 0.1 mg/l NAA as a rich source of phenolic acids, flavonoids, and catechins for pharmaceutical use, and/or health food production, and/or cosmological purposes. By using a precursor feeding strategy, it is possible to obtain even more valuable biomass from in vitro cultures, which is richer in antioxidants (1.33–2.33-fold) and may serve as the biotechnological raw material for the above-mentioned purposes. The next step in the biotechnological experiments should be the further optimization of the bioreactor culture conditions. It is possible at the same time to join PlantForm bioreactors and propose the production of *H. perforatum* cultivars biomasses on a commercial scale in the future.

## Figures and Tables

**Figure 1 molecules-28-02376-f001:**
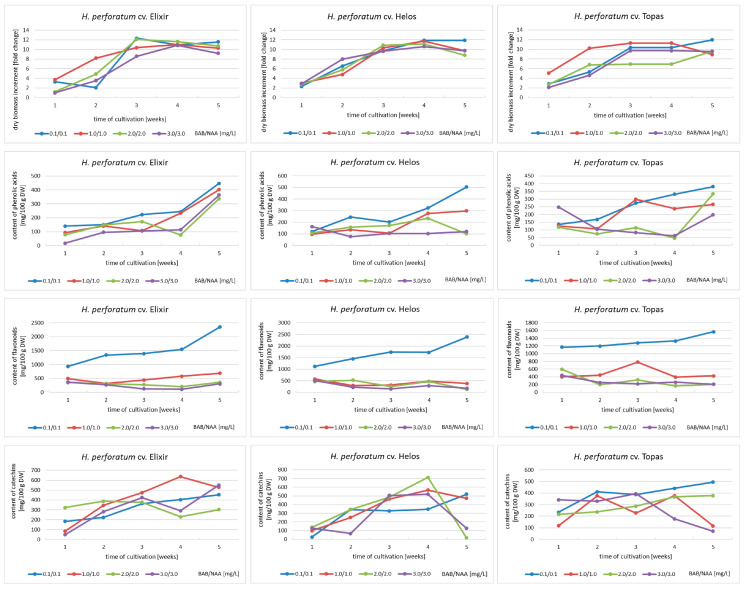
Dry biomass increments, and phenolic acid, flavonoid, and catechin contents in biomass extracts from agitated in vitro cultures of *H. perforatum* cultivars (Elixir, Helos, Topas) during 5-week growth cycles (MS medium variants with BAP and NAA). The metabolite content is expressed as the sum of the individual compounds determined by HPLC analysis.

**Figure 2 molecules-28-02376-f002:**
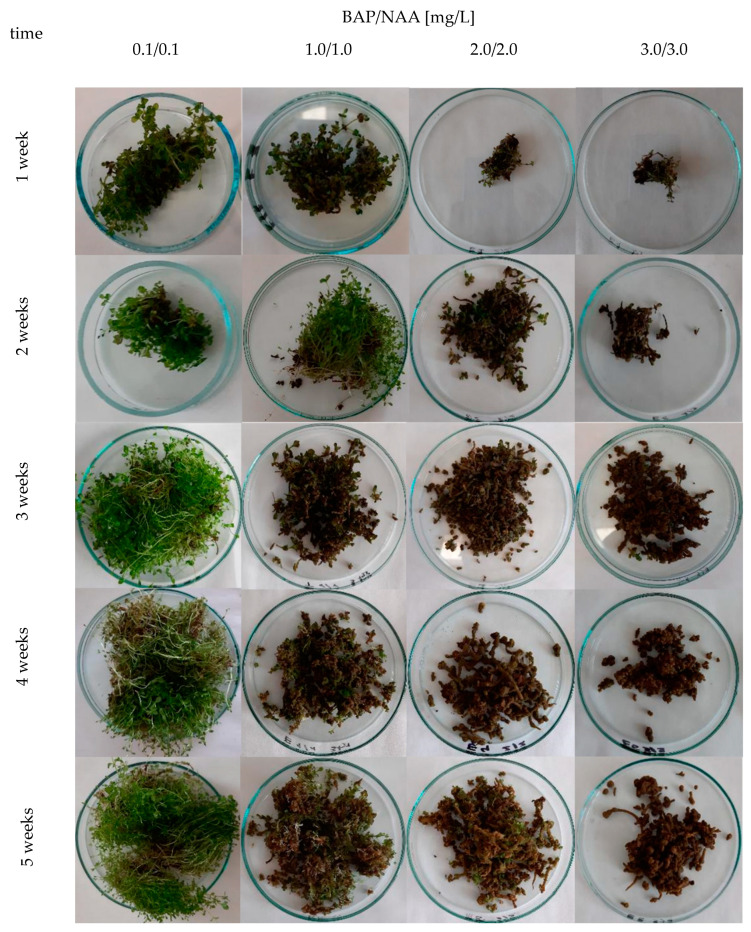
Morphology of microshoots from agitated *H. perforatum* cv. Elixir in vitro cultures depending on the concentration of PGRs tested (BAP and NAA) [mg/L] in a MS medium variant during 5-week growth cycles.

**Figure 3 molecules-28-02376-f003:**
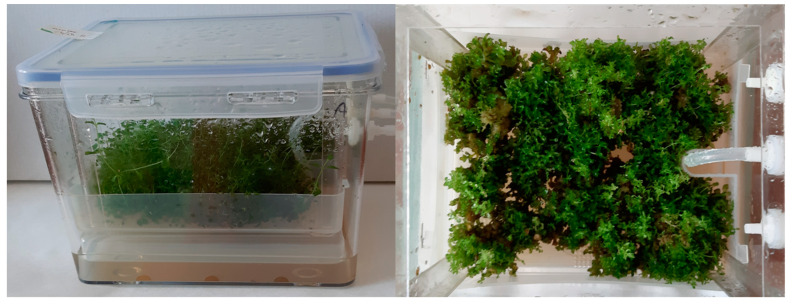
Microshoot in vitro culture of *H. perforatum* cv. Helos in the temporary immersion system (TIS): PlantForm bioreactors after a four-week growth cycle in a MS medium variant supplemented with 1.0 mg/L BAP and 1.0 mg/L NAA.

**Figure 4 molecules-28-02376-f004:**
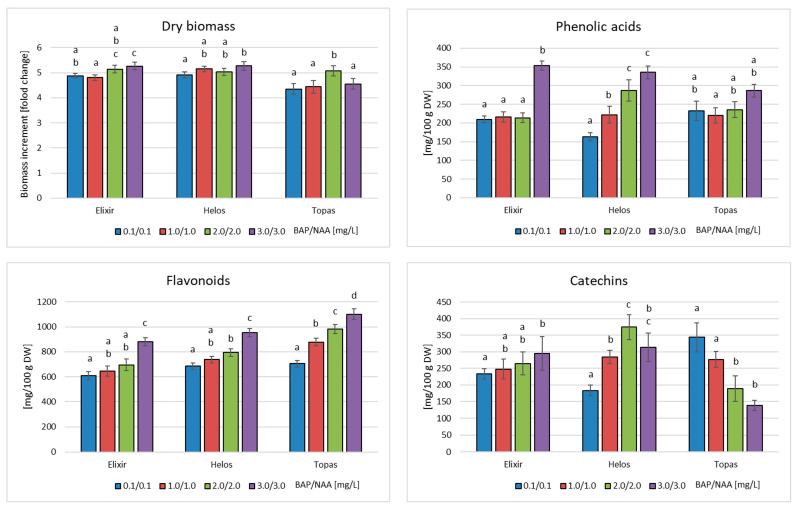
Dry biomass increments, and phenolic acid, flavonoid, and catechin contents in biomass extracts from in vitro cultures of *H. perforatum* cultivars (Elixir, Helos, Topas) after 4-week growth cycles in PlantForm bioreactors in MS medium with different amounts of PGRs. The metabolite content is expressed as the sum of the individual compounds determined by HPLC analysis. ^a–d^ Different letters indicate significant differences (*p* < 0.05).

**Table 1 molecules-28-02376-t001:** Content of total phenolics (calculated as gallic acid), total flavonoids (calculated as quercetin), and condensed tannins (calculated as catechin) in the methanolic extracts from the biomass of in vitro agitated cultures of the *H. perforatum* cultivars Helos, Elixir, and Topas (MS, 0.1 mg/L BAP and 0.1 mg/L NAA, 5 weeks).

*H. perforatum* Extracts	Polyphenol Content mg GAE/g Extract (DW)	Flavonoid Content mg QE/g Extract (DW)	Condensed Tannins mg CE/g Extract (DW)
cv. Helos	75.99 ± 0.19 ^a^	23.21 ± 0.29 ^a^	19.85 ± 0.36 ^a^
cv. Elixir	69.06 ± 1.17 ^b^	26.86 ± 0.22 ^a^	6.21 ± 0.23 ^b^
cv. Topas	75.49 ± 0.34 ^a^	48.33 ± 0.27 ^b^	16.31 ± 0.96 ^a^

Values are expressed as the mean ± SD (*n* = 3). ^a–b^ Different letters within the same column indicate significant differences between mean values (*p* < 0.05).

**Table 2 molecules-28-02376-t002:** Free radical scavenging activity (DPPH test), reducing power, ferrous ion (Fe^2+^) chelating activity of methanolic extracts obtained from the biomass of in vitro agitated cultures of the *H. perforatum* cultivars Helos, Elixir, and Topas (MS, 0.1 mg/L BAP and 0.1 mg/L NAA, 5 weeks).

*H. perforatum* Extracts	DPPH Test IC50 (mg/mL)	Reducing Power Assay ASE/mL	Fe^2+^ Chelating Activity IC_50_ (mg/mL)
cv. Helos	0.586 ± 0.062 ^a^	16.429 ± 0.229 ^a^	0.492 ± 0.012 ^a^
cv. Elixir	1.625 ± 0.048 ^b^	12.942 ± 0.839 ^b^	0.530 ± 0.100 ^b^
cv. Topas	0.512 ± 0.036 ^c^	20.765 ± 0.001 ^c^	0.505 ± 0.015 ^b^
Standard	BHT 0.0656 ± 0.008 ^d^	BHT 1.131 ± 0.037 ^d^	EDTA 0.0067 ± 0.0003 ^c^

Values are expressed as the mean ± SD (*n* = 3). ^a–d^ Different letters within the same column indicate significant differences between mean values (*p* < 0.05).

**Table 3 molecules-28-02376-t003:** Antibacterial activity (MIC—minimum inhibitory concentration) of the methanolic extracts obtained from the biomass of in vitro agitated cultures of the *H. perforatum* cultivars Helos, Elixir, and Topas against Gram-positive and Gram-negative bacteria (MS, 0.1 mg/L BAP and 0.1 mg/L NAA, 5 weeks).

Species of Microorganisms	Antibacterial Activity (MIC, mg/mL)		
	*H. perforatum* cv. Elixir	*H. perforatum* cv. Helos	*H. perforatum* cv. Topas
*Staphylococcus aureus*	15.0	7.5	10.0
*Staphylococcus epidermidis*	15.0	2.5	5.0
*Enterococcus faecalis*	15.0	10.0	10.0
*Enterococcus faecium*	15.0	10.0	10.0
*Bacillus subtilis*	10.0	2.5	2.5
*Escherichia coli*	40.0	40.0	40.0
*Enterobacter aerogenes*	40.0	40.0	40.0
*Enterobacter cloacae*	30.0	10.0	20.0
*Klebsiella pneumoniae*	40.0	30.0	40.0
*Pseudomonas aeruginosa*	30.0	20.0	30.0

**Table 4 molecules-28-02376-t004:** Antifungal activity (MIC—minimum inhibitory concentration) of the methanolic extracts obtained from the biomass of in vitro agitated cultures of the *H. perforatum* cultivars Helos, Elixir, and Topas (MS, 0.1 mg/L BAP and 0.1 mg/L NAA, 5 weeks).

Species of Microorganisms	Antifungal Activity (MIC, mg/mL)		
	*H. perforatum* cv. Elixir	*H. perforatum* cv. Helos	*H. perforatum* cv. Topas
*Candida albicans*	7.5	15.0	10.0
*Candida krusei*	7.5	30.0	20.0
*Candida quilliermondii*	30.0	30.0	30.0
*Aspergillus flavus*	30.0	40.0	40.0
*Penicillium chrysogenum*	15.0	30.0	30.0
*Trichophyton tonsurans*	7.5	2.5	2.5

**Table 5 molecules-28-02376-t005:** Metabolite content in the biomass extracts from agitated in vitro cultures of three *H. perforatum* cultivars supplemented with phenylalanine 1 g/L (MS, 0.1 mg/L BAP and 0.1 mg/L NAA). Values were determined on days 2, 4, and 7 after the addition of the precursor.

Metabolites [mg/100 g DW] ^1^		*Hypericum perforatum*					
		cv. Elixir		cv. Helos		cv. Topas	
	Day	Control	Phe	Control	Phe	Control	Phe
Protocatechuic acid	2	32.52 ^a^	26.55 ^a^	43.93 ^abcd^	38.79 ^abc^	41.51 ^abcd^	39.10 ^abc^
	4	61.60 ^cde^	58.65 ^bcde^	65.03 ^de^	64.94 ^de^	49.12 ^abcde^	31.63 ^a^
	7	68.97 ^e^	71.41 ^e^	72.14 ^e^	64.22 ^de^	57.26 ^bcde^	35.82 ^ab^
Neochlorogenic acid	2	118.75 ^a^	118.19 ^a^	137.25 ^abc^	115.53 ^abc^	138.67 ^a^	123.06 ^a^
	4	130.16 ^abc^	125.81 ^ab^	159.96 ^bc^	119.56 ^a^	122.30 ^a^	122.78 ^a^
	7	108.04 ^a^	120.11 ^a^	162.35 ^c^	131.43 ^abc^	124.18 ^a^	119.51 ^ab^
3,4-Dihydroxyphenylacetic acid	2	21.94 ^a^	35.76 ^a^	29.54 ^a^	25.93 ^a^	29.57 ^a^	48.34 ^ab^
	4	177.65 ^de^	126.02 ^c^	127.06 ^c^	128.05 ^c^	82.90 ^b^	158.94 ^cd^
	7	192.39 ^def^	264.95 ^g^	222.31 ^f^	170.06 ^de^	129.14 ^c^	201.26 ^ef^
Chlorogenic acid	2	2.55 ^a^	6.99 ^defg^	2.97 ^abc^	5.56 ^cdef^	3.16 ^abc^	2.83 ^a^
	4	4.08 ^abc^	4.89 ^abcd^	2.92 ^ab^	7.78 ^fg^	5.10 ^abcde^	4.58 ^abcd^
	7	5.50 ^bcdef^	15.56 ^h^	4.64 ^abcd^	7.68 ^efg^	3.88 ^abc^	8.33 ^g^
Cryptochlorogenic acid	2	47.37 ^cdef^	20.10 ^ab^	45.18 ^cde^	33.37 ^bcd^	61.32 ^efgh^	53.16 ^defg^
	4	69.90 ^ghij^	67.64 ^fghi^	78.75 ^hij^	56.69 ^efg^	72.29 ^ghij^	29.99 ^abc^
	7	71.50 ^ghij^	90.34 ^j^	83.02 ^ij^	44.75 ^cde^	65.95 ^efghi^	10.50 ^a^
*p*-Hydroxybenzoic acid	2	0.00 ^a^	120.46 ^bc^	0.00 ^a^	106.15 ^b^	0.00 ^a^	131.15 ^bc^
	4	0.00 ^a^	129.16 ^bc^	0.00 ^a^	141.78 ^cd^	0.00 ^a^	123.47 ^bc^
	7	0.00 ^a^	139.10 ^cd^	0.00 ^a^	164.98 ^d^	0.00 ^a^	162.49 ^d^
Vanillic acid	2	0.01 ^a^	0.01 ^a^	0.19 ^bcd^	0.24 ^cde^	0.01 ^a^	0.34 ^ef^
	4	0.01 ^a^	0.07 ^ab^	0.34 ^ef^	0.31 ^de^	0.29 ^de^	0.31 ^de^
	7	0.01 ^a^	0.12 ^abc^	0.82 ^gh^	0.44 ^f^	0.71 ^g^	0.85 ^h^
*p*-Coumaric acid	2	0.00 ^a^	59.47 ^bcd^	0.00 ^a^	45.48 ^b^	0.00 ^a^	56.24 ^bc^
	4	0.00 ^a^	67.12 ^bcd^	0.00 ^a^	71.29 ^cd^	0.00 ^a^	64.22 ^bcd^
	7	0.00 ^a^	69.50 ^cd^	0.00 ^a^	78.15 ^d^	0.00 ^a^	74.36 ^cd^
Total phenolic acids	2	223.14 ^a^	387.53 ^bc^	259.06 ^ab^	371.05 ^bc^	274.24 ^ab^	454.22 ^cde^
	4	443.39 ^cd^	579.35 ^def^	434.04 ^cd^	590.40 ^ef^	332.00 ^abc^	535.91 ^def^
	7	446.40 ^cde^	771.09 ^g^	545.27 ^def^	661.69 ^fg^	381.12 ^bc^	613.12 ^f^
Hyperoside	2	3.84 ^a^	145.29 ^cd^	9.79 ^a^	110.52 ^b^	8.85 ^a^	94.84 ^b^
	4	6.56 ^a^	117.42 ^b^	6.37 ^a^	119.94 ^bc^	6.82 ^a^	113.56 ^b^
	7	11.61 ^a^	220.39 ^e^	9.71 ^a^	158.03 ^d^	7.89 ^a^	155.96 ^d^
Rutoside	2	41.80 ^efg^	53.63 ^fgh^	45.01 ^efgh^	18.45 ^abc^	54.99 ^fgh^	13.09 ^ab^
	4	65.17 ^hi^	39.61 ^def^	59.18 ^fgh^	31.66 ^cde^	53.14 ^fgh^	62.29 ^ghi^
	7	97.56 ^j^	91.48 ^j^	6.60 ^a^	131.31 ^k^	56.22 ^fgh^	83.25 ^ij^
Apigetrin	2	0.00 ^a^	59.76 ^bc^	0.00 ^a^	45.67 ^b^	0.00 ^a^	42.54 ^b^
	4	0.00 ^a^	75.67 ^cd^	0.00 ^a^	83.46 ^de^	0.00 ^a^	75.36 ^cd^
	7	0.00 ^a^	97.34 ^ef^	0.00 ^a^	115.64 ^fg^	0.00 ^a^	120.47 ^g^
Quercitrin	2	62.57 ^c^	56.74 ^bc^	66.51 ^c^	28.32 ^ab^	57.02 ^bc^	17.30 ^a^
	4	69.91 ^c^	166.08 ^fg^	54.71 ^bc^	151.34 ^ef^	55.07 ^bc^	135.10 ^e^
	7	102.60 ^d^	196.36 ^h^	9.27 ^a^	184.63 ^gh^	60.18 ^c^	126.36 ^de^
Quercetin	2	1396.34 ^abcd^	1237.51 ^ab^	1606.27 ^cde^	1582.11 ^cde^	1175.05 ^a^	1776.04 ^e^
	4	1729.13 ^e^	1700.44 ^de^	1541.45 ^bcde^	1522.16 ^bcde^	1184.85 ^a^	1726.70 ^e^
	7	2233.06 ^f^	2431.22 ^f^	2335.43 ^f^	2258.23 ^f^	1335.28 ^abc^	2796.61 ^g^
Luteolin	2	9.89 ^ab^	69.33 ^g^	4.47 ^a^	29.45 ^bcde^	12.63 ^abc^	58.70 ^g^
	4	15.14 ^abcd^	37.54 ^ef^	10.58 ^ab^	33.35 ^cdef^	17.86 ^abcde^	33.79 ^def^
	7	13.33 ^abcd^	27.72 ^bcde^	15.03 ^abcd^	33.21 ^cdef^	17.53 ^abcde^	51.35 ^fg^
Kaempferol	2	7.30 ^a^	14.09 ^abcd^	8.16 ^a^	13.88 ^abcd^	9.10 ^a^	10.60 ^ab^
	4	10.24 ^a^	19.98 ^cd^	9.60 ^ab^	19.26 ^bcd^	9.09 ^a^	11.47 ^abc^
	7	9.68 ^a^	21.54 ^d^	9.63 ^a^	5.57 ^a^	22.66 ^d^	10.72 ^abc^
Total flavonoids	2	1521.74 ^ab^	1636.35 ^abc^	1740.21 ^abcd^	1828.41 ^bcd^	1317.65 ^a^	2013.11 ^cde^
	4	1896.15 ^bcd^	2156.73 ^def^	1681.89 ^abc^	1961.16 ^cde^	1326.83 ^a^	2158.28 ^def^
	7	2467.83 ^fg^	3086.05 ^hi^	2385.67 ^ef^	2886.63 ^gh^	1499.76 ^ab^	3344.73 ^i^
Epigallocatechin	2	0.00 ^a^	20.54 ^bc^	0.00 ^a^	29.54 ^d^	0.00 ^a^	15.15 ^b^
	4	0.00 ^a^	29.15 ^cd^	0.00 ^a^	31.54 ^d^	0.00 ^a^	16.45 ^b^
	7	0.00 ^a^	36.05 ^d^	0.00 ^a^	33.48 ^d^	0.00 ^a^	20.47 ^b^
Catechin	2	220.80 ^e^	112.03 ^bc^	176.16 ^d^	138.98 ^c^	224.56 ^ef^	86.71 ^b^
	4	227.90 ^ef^	175.12 ^d^	258.36 ^f^	139.39 ^c^	232.45 ^ef^	124.30 ^c^
	7	214.26 ^e^	120.54 ^bc^	311.68 ^g^	21.10 ^a^	236.07 ^ef^	15.59 ^a^
Epicatechin	2	182.97 ^cd^	361.71 ^g^	149.76 ^bc^	324.94 ^g^	218.56 ^def^	141.44 ^ab^
	4	223.19 ^def^	460.78 ^i^	217.64 ^def^	428.10 ^hi^	209.20 ^de^	253.56 ^f^
	7	253.97 ^f^	407.47 ^h^	195.33 ^d^	450.95 ^i^	244.60 ^ef^	101.25 ^a^
Epicatechin gallate	2	0.00 ^a^	45.04 ^de^	0.00 ^a^	58.14 ^fg^	0.00 ^a^	23.16 ^b^
	4	0.00 ^a^	51.54 ^ef^	0.00 ^a^	60.50 ^fg^	0.00 ^a^	29.46 ^bc^
	7	0.00 ^a^	58.25 ^fg^	0.00 ^a^	67.24 ^g^	0.00 ^a^	35.41 ^cd^
Total catechins	2	403.76 ^cd^	539.32 ^fghi^	325.92 ^bc^	551.60 ^ghi^	443.12 ^de^	266.46 ^b^
	4	451.09 ^de^	716.58 ^k^	475.99 ^defg^	659.53 ^jk^	441.65 ^de^	423.77 ^d^
	7	468.23 ^def^	622.31 ^ij^	507.01 ^efgh^	572.77 ^hi^	480.67 ^defg^	172.72 ^a^

^1^ The same letter indicates no significant differences between mean values (*p* < 0.05).

**Table 6 molecules-28-02376-t006:** Microbial strains used in antimicrobial tests of the methanol extracts from the biomass of in vitro cultures of the *H. perforatum* cultivars Helos, Elixir, and Topas.

Groups of Microorganisms	Name and Number of Strains
Gram-positive bacteria	1. *Staphylococcus aureus* ATCC 6538 P
	2. *Staphylococcus epidermidis* S3
	3. *Enterococcus faecalis* ATCC 8040/1
	4. *Enterococcus faecium* 34B/8
	5. *Bacillus subtilis* ATCC 6633
Gram-negative bacteria	6. *Escherichia coli* ATCC 8739
	7. *Enterobacter aerogenes* 35B
	8. *Enterobacter cloacae* 382/2
	9. *Klebsiella pneumoniae* ATCC 16903
	10. *Pseudomonas aeruginosa* ATCC 27853
Yeast-like fungi	11. *Candida albicans* PCM 1409 PZH
	12. *Candida krusei* S220
	13. *Candida quilliermondii* 11
Molds	14. *Aspergillus flavus* 35/1
	15. *Penicillium chrysogenum* ATCC 10106
Dermatophytes	16. *Trichophyton tonsurans* 2

## Data Availability

Not applicable.

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
