# Peer review of "Different Types of Hypericum perforatum cvs. (Elixir, Helos, Topas) In Vitro Cultures: A Rich Source of Bioactive Metabolites and Biological Activities of Biomass Extracts"

_molecules, 2023, doi:10.3390/molecules28052376_

Round 1

Reviewer 1 Report

Please find my comment in the attached file

Reviewer 2 Report

The current study was based on Microshoot agitated and bioreactor cultures method, using four variants of Murashige and Skoog medium (MS) supplemented with different concentrations 6-benzylaminopurine (BAP) and 1-naphthaleneacetic acid (NAA) (in the range of 0.1-3.0 mg/L) to culture three Hypericum perforatum cvs (Elixir, Helos, Topas), and compared the contents of active substances in different growth stages. In addition, the bioactivities (antioxidant activity; antimicrobial activity, antifungal activity) of its extracts were evaluated. The biomass extracts showed high amounts of phenolic acids, flavonoids, and catechins and  high or moderate antioxidant activity, high activity against Gram-positive bacteria, and strong antifungal activity.

Several issues as follows:

1: Page1: Lines 25: “The highest total contents of phenolic acids, flavonoids, and catechins were 505, 2386, and 712 mg/100 g DW-, respectively (agitated cultures of cv. Helos). Pay attention to the correct writing of “DW-”.

2: Page21. Introduction Lines 52-55:The herb of H. perforatum exhibits antidepressant properties and is one of the top-selling antidepressants in the world. It also shows astringent and spasmolytic effects, and it accelerates the healing of smaller wounds and burns. The main metabolites exhibit antimicrobial activity. Here mentioned antidepressant properties, astringent and spasmolytic effects, please supplement the references.

3. Page 4: Lines 159-161: Here it turns out thatFor the Elixir and Helos cultivars, the maximum content of this group of metabolites was achieved in the fourth week of cultivation. For the Topas cultivar, it was the fifth week.” Why was Condition “MS, 0.1 mg/L BAP and 0.1 mg/L NAA, 5 weeks” used in later tests.

4. Page 10: Lines 361-362:Both cultivars Topas and Helos displayed good radical scavenging properties (IC50 0.512 ± 0.036 mg/mL and 0.586± 0.062 respectively) (Table 2).” Please substitute IC50= 0.512 ± 0.036 mg/mL and 0.586± 0.062 for IC50 0.512 ± 0.036 mg/mL and 0.586± 0.062.

5. Page 10: Lines 393: “several compounds in these classes have been shown to possess powerful antioxidant activity in both in vitro and in vivo investigations [37].” Adding several representative compounds with antioxidant activity makes the article more convincing.

6. Page18:  “3.5. Extracts preparation”: Please add how is the methanol extract stored?

Reviewer 3 Report

The proposed article is of great interest, it is well written and the results support its objective.
However, I have a question and it is related to the statistical study. In the tables shown it is not seen that any statistical test has been carried out even though the study has been carried out with replicates, could they include it?

Round 2

Reviewer 3 Report

Accept in present form